# Interferon Lambda Delays the Emergence of Influenza Virus Resistance to Oseltamivir

**DOI:** 10.3390/microorganisms9061196

**Published:** 2021-06-01

**Authors:** Chiara Medaglia, Arnaud Charles-Antoine Zwygart, Paulo Jacob Silva, Samuel Constant, Song Huang, Francesco Stellacci, Caroline Tapparel

**Affiliations:** 1Department of Microbiology and Molecular Medicine, University of Geneva, 1206 Geneva, Switzerland; chiara.medaglia@unige.ch (C.M.); arnaud.zwygart@unige.ch (A.C.-A.Z.); 2Insitute of Materials, Ecole polytechnique fédérale de Lausanne, 1015 Lausanne, Switzerland; paulo.jacob@epfl.ch (P.J.S.); francesco.stellacci@epfl.ch (F.S.); 3Epithelix Sas, 1228 Geneva, Switzerland; samuel.constant@epithelix.com (S.C.); song.huang@epithelix.com (S.H.)

**Keywords:** influenza virus, oseltamivir, interferon lambda, neuraminidase, antiviral resistance

## Abstract

Influenza viruses are a leading cause of morbidity and mortality worldwide. These air-borne pathogens are able to cross the species barrier, leading to regular seasonal epidemics and sporadic pandemics. Influenza viruses also possess a high genetic variability, which allows for the acquisition of resistance mutations to antivirals. Combination therapies with two or more drugs targeting different mechanisms of viral replication have been considered an advantageous option to not only enhance the effectiveness of the individual treatments, but also reduce the likelihood of resistance emergence. Using an in vitro infection model, we assessed the barrier to viral resistance of a combination therapy with the neuraminidase inhibitor oseltamivir and human interferon lambda against the pandemic H1N1 A/Netherlands/602/2009 (H1N1pdm09) virus. We serially passaged the virus in a cell line derived from human bronchial epithelial cells in the presence or absence of increasing concentrations of oseltamivir alone or oseltamivir plus interferon lambda. While the treatment with oseltamivir alone quickly induced the emergence of antiviral resistance through a single mutation in the neuraminidase gene, the co-administration of interferon lambda delayed the emergence of drug-resistant influenza virus variants. Our results suggest a possible clinical application of interferon lambda in combination with oseltamivir to treat influenza.

## 1. Introduction

Influenza is an infectious respiratory disease caused in humans by influenza A (IAV) and influenza B (IBV) viruses. It affects approximately 1 billion individuals each year and, according to the World Health Organization, the annual mortality burden of this disease spans between 250,000 and 500,000 deaths worldwide [1]. Besides the annual seasonal epidemics, more rare and unpredictable pandemic outbreaks that involve IAVs of zoonotic origin also represent a threat. IAVs cause pandemics when they acquire the ability to infect and transmit between different species, generating an antigenically novel virus [2]. In the past hundred years, four influenza pandemics occurred, all associated with higher mortality rates than seasonal epidemics [3]. In this time frame, globalization has driven social and economic changes that have enhanced the threat of disease emergence and accelerated the spread of novel strains. Due to their error-prone polymerase, IVs rapidly acquire genetic variability that inevitably culminates in the emergence of resistance to antivirals. Even changes in a very small number of amino acid residues in the targeted viral protein can be sufficient to reduce or completely block the efficacy of a drug [4]. Importantly, antiviral resistance may not necessarily result from the drug selective pressure, but it can also develop in the absence of treatments [5].

The neuraminidase inhibitors (NAIs) oseltamivir, peramivir, zanamivir, and laninamivir are currently the first-line treatment for both IAV and IBV. They competitively inhibit neuraminidase (NA) on the surface of newly formed viral particles, thus preventing their shedding from the host cells. One strength of the NAIs over the older adamantanes is that they are less prone to select for resistant mutants [6,7]. Therefore, the emergence of resistant variants against this class of antivirals, about 15 years ago, was a cause of immediate concern [8,9]. Oseltamivir (OS), sold under the brand name Tamiflu^TM^, is taken per os and used for both the treatment and the prophylaxis of influenza. Among the NAIs, OS is certainly the most commonly used because it can easily be self-administered [10]. A single H275Y amino acid substitution (H274Y in N2 numbering) in the IAV NA gene confers resistance to OS [11,12]. This substitution emerged first in the seasonal influenza A H1N1 (A/Brisbane/59/2007-like) strain in 2007 and rapidly spread to all H1N1 strains [13]. The 2009 H1N1pdm09 was fortunately sensitive to the drug although clusters of OS-resistant (OR) strains are detected at low frequency (<2%) [14,15,16,17,18], posing a threat of global spread of resistance as occurred with the seasonal pre-pandemic H1N1. In influenza wild-type strains, the NA active site changes shape to accommodate OS. H275Y inhibits the binding of the drug by preventing this conformational change. This mutation reduces the susceptibility of H1N1 IAV to OS by approximately 400-fold [19]. Moreover, due to the acquisition of additional permissive mutations preserving viral fitness, the H275Y variants have been shown to persist even after cessation of the treatment, with a morbidity and mortality profile similar to their wild-type counterparts [12,20]. Similar resistance problems are encountered when using the other anti-IV antivirals, including the most recent ones, targeting the RNA-dependent RNA polymerase (RdRp) [21]. Thus, there is an unmet need for new treatment regimens that can reduce the risk of resistance appearance.

Combination therapy is considered to be a valuable approach to limit the emergence of drug resistance. The rationale behind this concept is that while IV can rapidly develop resistance to a single antiviral, it takes longer to develop resistance to two or more drugs simultaneously [22]. In line with that, combining drugs targeting different mechanisms of viral replication may be more effective in decreasing the emergence of resistance, as demonstrated when amantadine and ribavirin were combined with OS in an in vivo mouse model of influenza [23]. Given that scenario, we chose to optimize the use of OS and to evaluate its propensity to select for resistant mutants when administered alone or together with human interferon lambda 1 (IFN λ1).

Interferons (IFNs) are a class of innate cytokines produced in response to viral infection. IFNs type I and type III (alias IFN λ) are the host frontline defense against viruses. Once they bind to their receptors, IFNs activate a gene expression program that induces an antiviral state and limits the spread of the infection [24]. IFN type I is an FDA-approved drug [25]. However, as it is able to directly activate immune effector cells, IFN type I can trigger a massive immune response that may exacerbate the outcome of influenza infection [26]. On the other hand, IFN λ, thanks to its restricted receptor distribution, only acts on the epithelial barriers, without causing the adverse inflammatory effects associated with IFN type I [27,28]. These properties suggest IFN λ as a treatment of choice against viral infections, with a higher tolerability than IFN type I. Several studies demonstrate that IFN λ enhances the adaptive immune response in the respiratory mucosa, without compromising the host fitness [29,30]. IFN λ was also found to play a critical early role, not shared by IFN type I, in protection of the lung following influenza virus infection [31,32]. It is reported that IFN λ exerts variable degrees of antiviral activity in vivo without emergence of resistance against both IV viruses [33]. Moreover, human IFN λ1 has been successfully used in clinical trials against hepatitis C [33,34,35,36,37].

Previous studies have proved that OS and IFN λ1 display synergistic antiviral activity against the H1N1 IAV [38] but, to the best of our knowledge, their combined effect on the emergence of antiviral resistance has never been addressed. Hence, we asked whether the co-administration of these two classes of anti-influenza drugs could alter the emergence of resistant variants. We used Calu-3 cells, derived from human bronchial submucosal glands [39], to serially passage H1N1pdm09 influenza virus in the presence of increasing concentrations of OS, IFN λ1, OS plus IFN λ1, or medium alone to generate viral variants that displayed various degrees of antiviral resistance. Calu-3 cells effectively support IV infection and represent a reliable model to evaluate the emergence of NAI resistant strains, as well as the interaction between OS and IFN λ1.

## 2. Materials and Methods

### 2.1. Cells, Tissues, Viruses, and Compounds

Madin–Darby canine kidney (MDCK) and Calu-3 cell lines were obtained from the American Type Culture Collection (Manassas, VA, USA). Calu-3 cells were cultured in Minimum Essential Medium (MEM) supplemented with GlutaMAX^TM^, 10% FBS, Phenol Red, 1% Hepes, 1% Non-Essential Amino Acids, 1% penicillin/streptomycin, and 1% Sodium-pyruvate and grown at 37 °C in an atmosphere of 5% CO2. MDCK cells were cultured in Dulbecco’s Modified Eagle Medium (DMEM) supplemented with GlutaMAX^TM^, Sodium Pyruvate, Phenol Red, 10% FBS, and 1% P/S and grown at 37 °C in an atmosphere of 5% CO2. Human ex vivo reconstituted upper respiratory tissues, Mucilair, were purchased from Epithelix (Geneva, Switzerland) and maintained in an air–liquid interface according to the manufacturer’s instructions [40].

Human recombinant IFN λ1 protein was obtained from R&D Systems, Inc. (Abingdon, United Kingdom). OS carboxylate was provided by Roche Diagnostics GmbH (Mannheim, Germany).

Human H1N1, A/Netherlands/602/2009 influenza virus (A(H1N1)pdm09), kindly provided by Prof. Mirco Schmolke (University of Geneva), was amplified and titrated in MDCK cells by plaque assay. For viral stock production, the cells were infected with a multiplicity of infection (MOI) of 0.01 PFU/cell in serum-free DMEM for 1 h at 37 °C. The inoculum was then removed and fresh serum-free medium containing 1 μg/mL of TPCK trypsin was added. The infectious supernatant was collected 48 h post infection (hpi), aliquoted, and frozen at −80 °C before titration. Viral stocks of A(H1N1)pdm09 variants were prepared in Calu-3 cells infected with a MOI of 0.1 PFU/cell, in serum-free MEM for 1 h at 37 °C. Upon inoculum removal, fresh serum-free medium was added and the infectious supernatant was collected at 48 hpi, aliquoted, and frozen at −80 °C before titration in MDCK cells. All experimental work was performed in a biosafety level 2 laboratory approved for use of these strains.

### 2.2. Cell Viability Assay

Calu-3 cells (1 × 10^5^ cells per well) were seeded in a 96-well plate one day before the assay. A dose range of IFN λ1 (spanning from 125 ng/mL to 1 μg/mL), OS (spanning from 160 to 640 μM), or IFN λ1 plus OS was added to the cells in serum-free MEM for 48 h. MTT reagent (Promega) was added to the cells for 3 h at 37 °C according to the manufacturer’s instructions. Subsequently, the absorbance was read at 570 nm. Percentages of viability were calculated by comparing the absorbance in treated wells and untreated conditions.

### 2.3. Infectivity of A(H1N1)pdm09 Influenza Viruses Measured by Plaque Assay in MDCK

The infectivity of the WT virus and the selected variants was determined by at least two independent plaque assays. Briefly, confluent cultures of MDCK cells in 6 multiwell plates were incubated at 37 °C for 1 h with 10-fold serial dilutions of each virus prepared in serum-free DMEM containing 1% penicillin/streptomycin. Upon inoculum removal, the cells were washed and overlaid with MEM containing 0.3% BSA, 0.9% Bacto agar, and 1 μg/mL TPCK-treated trypsin. After 48 h of incubation at 37 °C, the cells were fixed with 4% formaldehyde solution and then stained with 0.1% crystal violet. The number of PFUs per dilution was determined using a fine scale magnifying comparator and a white light table.

### 2.4. Virus Yield Reduction Assay in Calu-3 Cells

Confluent layers of Calu-3 cells seeded in 96-well plates were infected with a MOI of 0.1 PFU/cell of the original viral stock or with the four selected variants (i.e., OS p9, λ p9, OS/λ p9, and UTR p9) in serum-free MEM for 1 h at 37 °C.

A dose range of IFN λ1 spanning from 12.5 ng/mL to 100 ng/mL was added to the cells for 24 h before infection. One hour post infection, the inoculum was removed and the same dose range of drug was added to the cells. A dose range of OS spanning from 1.8 μM to 15 μM was added only at 1 hpi and the cells were further incubated in a drug-containing medium for 24 h. Then, the supernatant was collected and infectious virus yields were determined as the number of PFUs/mL in MDCK cells. The drug concentration that caused a 50% decrease in the PFU titer in comparison to control wells without drug was defined as the half maximal effective concentration (EC50). The results of two independent experiments, each consisting of two replicates, were averaged. The EC50 values were calculated using Prism 8.0 (GraphPad, San Diego, CA, USA).

### 2.5. Selection of Resistant Variants

A(H1N1)pdm09 influenza viruses were successively passaged with a MOI of 0.1 PFU/cell in Calu-3 cells, seeded in 6-well plates, in the presence or absence of OS, IFN λ, OS plus IFN λ, or serum-free MEM only. The first administered dose of each compound, 10 µM for OS and 37 ng/mL for IFN λ, was doubled at each passage until the toxic dose was reached. Human IFN λ1 was administered 24 h before the infection (hbi) and then the same dose of drug was added to the cells at 1 hpi, right after the inoculum removal. OS was always administered 1 hpi, in both single and combined treatments. The cells were thus incubated in a drug containing medium for 48 h. Then, the supernatants were collected and centrifuged at 3000 rpm for 5 min in order to separate the dead cells from the viral suspensions. The supernatants were aliquoted and stored at −80 °C before being titrated in MDCK. Infectious virus yields were determined as the number of PFU/mL in MDCK cells. The *p* values were calculated on logarithmic values using the two-way ANOVA with Prism 8.0 (GraphPad, San Diego, CA, USA).

### 2.6. Assessment of Viral Fitness in Calu-3 and in Respiratory Tissues

To determine viral fitness in vitro, Calu-3 cells were infected with the A(H1N1)pdm09 viruses (the original viral stock or the four selected variants) at a MOI of 0.01 PFU/cell. After incubation for 1 h, the cells were washed and overlaid with serum-free MEM containing 1% penicillin/streptomycin. The infectious supernatants were collected at 24, 48, and 72 hpi and stored at −80 °C until titration in MDCK cells. At each time point, the entire supernatant was collected and new medium was added, thus allowing for the measurement of daily viral production.

To determine viral fitness ex vivo, Mucilair tissues were infected on their apical side with 5 × 10^4^ PFU/tissue of the A(H1N1)pdm09 viruses. The viruses were diluted in Mucilair medium and the infection was performed at 33 °C for 4 h. The inocula were then removed and the tissues were washed 5 times with DPBS [40]. The viral particles released from the apical side of the tissues were recovered every 24 h for four days and then the number of RNA copies were obtained by RT-qPCR [40].

### 2.7. RT-qPCR Analysis and Viral RNA Copies Quantification

Viral RNA was extracted from Mucilair apical washes using an EZNA viral extraction kit (Omega Biotek, Norcross, GA, USA) and quantified by using RT-qPCR with the QuantiTect kit (#204443; Qiagen, Hilden, Germany) in a StepOne ABI Thermocycler. Viral RNA copies were quantified as follows: 4 ten-fold dilution series of in vitro transcripts of the influenza A/California/7/2009(H1N1) M gene were used as the reference standard as previously described [41]. CT values were converted into RNA load using the slope–intercept form. In all experiments, the slope, efficiency, and R2 ranged between 0.96 and 0.99 [40,42].

### 2.8. Virus Sequencing

Viral RNAs were isolated from virus-containing cell culture fluid after passages in Calu-3 cells. Samples were then reverse-transcribed. The RT reaction was performed with a mix composed of First Strand Buffer 5× (InvitrogenTM), H2O Rnase free, Superscript ^®^ III RT/Platinum^®^ Taq Mix (InvitrogenTM), 0.1 M DTT (InvitrogenTM), dNTPs (25 mM), Protector RNase inhibitor (40 U/µL) (Roche), and Random hexamers (50 µg/µL) (InvitrogenTM).

Viral gene segments were then amplified from the viral cDNA using a master mix composed of 10× PCR Rxn buffer (-MgCl2) (InvitrogenTM), 50 mM MgCl2 (InvitrogenTM), H2O Rnase free, Platinum^®^ Taq DNA polymerase (5 U/µL) (InvitrogenTM), dNTPs (10 mM), and M13-tailed primers specific for the hemagglutinin (HA) and NA segments (Table 1). The length and the quality of the amplified fragments were verified by electrophoresis in an agarose gel 1% with 4 µL of Syber-safe and purified with an MSB Spin PCRapace column (Stratec, Berlin, Germany). The samples were sequenced through the Fasteris DNA sequencing service (Geneva) and then analyzed using the Geneious program [43]. For minority species analysis, an internal sequencing primer was used (Table 1, NA seq).

### 2.9. C11-6′ Inhibition Assay

The C11-6′ inhibition assay was performed as previously described [44]. A dose range of C11-6′ spanning from 1.2 μg/mL to 300 μg/mL was pre-incubated with 0.1 MOI of UTR p9 N09 Stock, OS p9, OS/λ p9, or λ p9 for 1 h in serum-free DMEM at 37 °C. The mix virus plus drug was then inoculated for 1 h at 37 °C on a confluent layer of MDCK cells seeded in a 96-well plate. The inocula were then removed and the cells were overlaid with serum-free DMEM containing 1% penicillin/streptomycin for 12 h at 37 °C. The number of infected cells was calculated by immunocytochemistry. After fixation in methanol, the primary antibody (mouse monoclonal influenza A antibody 1:100 dilution, Chemicon^®^) was added for 1 h at 37 °C. The cells were then washed with DPBS/Tween 0.05% three times and the secondary antibody (Anti-mouse IgG, HRP-linked 1:500 dilution, Cell signaling technology) was added. After 1 h, the cells were washed and the DAB solution was added. Infected cells were counted and percentages of infection were calculated by comparing the number of infected cells in treated and untreated conditions. All results are presented as the mean values from two independent experiments performed in duplicate. The EC50 values for inhibition curves were calculated by regression analysis using the program GraphPad Prism version 8.0 (GraphPad Software, San Diego, CA, USA).

## 3. Results

### 3.1. Determination of IFN λ1 and Oseltamivir Non-Toxic Doses

In order to assess the antiviral activity of IFN λ1 and OS, we first defined their non-toxic dose range by measuring their effect on Calu-3 cells’ metabolic activity by MTT Assay. Calu-3 cells are derived from human bronchial adenocarcinoma and are commonly used as an in vitro model for IV virus infection [39,45,46]. They express the TMPRSS2 protease needed for hemagglutinin (HA) maturation and therefore allow for several cycles of IV replication, with no need to add exogenous proteases such as trypsin in the cell culture medium [47]. We evaluated the toxic activity of individual treatments over 48 h (Figure 1A,B). We chose the dose range of each compound based on previously published data [38]. No toxicity was detected for any of the tested concentrations, as the reduction in cell viability never went below 20% compared with the untreated control. Then, we assessed the toxicity of IFN λ1 plus OS combined with similar dose ranges (Figure 1C). Also this time, no effect on the viability of Calu-3 cells was detected.

### 3.2. Susceptibility of A(H1N1)pdm09 Virus to Oseltamivir and IFN λ1

Next, we assessed the antiviral activity of both OS and IFN λ1 against A(H1N1)pdm09. A dose range of OS (spanning from 1.8 μM to 15 μM) was added to Calu-3 cells in a post-treatment, i.e., administered 1 h post infection (hpi), right after the inoculum removal. Unlike OS, IFN λ1 exerts an antiviral effect only when administered in a pre- plus a post-treatment in cell lines [38,48], as its antiviral effect relies on the activation of a gene expression program, which entails the need for a pre-treatment in vitro. Thus, a dose range of IFN λ1 (from 12.5 ng/mL to 100 ng/mL) was administered at both 24 h before the infection (hbi) and again at 1 hpi. The cells were then incubated in a drug-containing medium for 24 h. To quantify the efficacy of the treatments, the number of viral particles present in the supernatant of infected Calu-3 cells at 24 hpi was determined by plaque assay in MDCK cells. We therefore determined the percentage of infectivity by comparing the numbers of viral particles present in each experimental condition to that of the untreated infected control. The drug concentrations that caused either a 50% or a 90% decrease in the PFU titer in comparison with control wells without drug were defined as the half maximal effective concentration (EC50) or the effective concentration 90 (EC90), respectively (Figure 2A). Then, following the above-described administration protocol, we verified that IFN λ1, though poorly active per se, enhanced the effect of OS in our settings. Specifically, we tested the antiviral activity of the combined dose ranges of the two compounds shown in Figure 2A. At any of the tested combined treatments, the antiviral effect was greater than that of OS alone (Figure 2B), confirming a synergistic effect of the two compounds, as published [38].

### 3.3. Selection of Resistant Variants against Oseltamivir Administered Alone or in Combination with IFN λ1

IFN λ1 and OS display synergistic activity against H1N1 IAV in vitro [38]. However, the question of whether the co-administration of IFN λ1 could also increase the barrier to OS resistance has not been investigated yet. To evaluate the effects of the single and combined treatment with OS and IFN λ1 on the emergence of A(H1N1)pdm09 resistant variants, we serially passaged the virus 12 times in Calu-3 cells [49,50], in the presence of increasing amounts of OS (10 μM to 640 μM), IFN λ1 (37 ng/mL to 1000 ng/mL), or OS plus IFN λ1 (Figure 3). The starting concentrations of 37 ng/mL for IFN λl and 10 µM of OS were used at passage 1 (p1) and were doubled at each passage. When the highest non-toxic concentration tested was reached, the dose of the compound was kept constant for the subsequent passages (Figure 3D). At each passage, the cells were infected with a MOI of 0.1 PFU/cell for each condition. Calu-3 cells were administered with IFN λ1 both 24 hbi and 1 hpi, while OS was given only at 1 hpi. At 48 hpi, the infectious viral titers and the plaque size were measured by plaque assay in MDCK cells. Of note, in order to monitor the effects of adaptation of A(H1N1)pdm09 to growth in Calu-3 cells, we also passaged the parental virus in parallel in the absence of treatments. Interestingly, we did not observe differences in plaque morphology across conditions at any passage.

Viruses passaged in the presence of OS alone produced titers significantly lower (viral yield reduction > 1 log) than those of the untreated control (UTR CTRL) at p2, p3, and p4. However, at p7 there was no more difference between the titers produced in the presence of OS alone and those generated in the UTR CTRL, suggesting the emergence of an OS-resistant variant (Figure 3A).

Viruses passaged in the presence of IFN λ1 alone produced titers not significantly different from those measured in the UTR CTRL (Figure 3B). This result was expected as, due to the lack of innate immune cells, the antiviral activity of IFN λ1 in vitro is weak [48], especially if assessed when the viral replication reached its peak (i.e., 48 hpi versus 24 hpi in Figure 2A).

On the other hand, viruses passaged in the presence of OS plus IFN λ1 produced titers always lower (except for P4) than those measured in the UTR CTRL, this up to p11. At p12, the treated viruses displayed the same viral titer as the UTR CTRL, indicating a reduced susceptibility to OS. Of note, the emergence of antiviral resistance in the OS condition was observed between p6 and p7 (Figure 3A), while in the OS + IFN λ1 condition the susceptibility of the viruses to OS was still evident at this passage, and lasted until p11 (Figure 3C). This finding strongly suggested that the co-administration of IFN λ1 delays the emergence of antiviral resistance to OS.

To confirm the resistance phenotype of the A(H1N1)pdm09 variants, we measured their degree of susceptibility to OS and IFN λ1. We chose the viruses collected after nine passages (as at p9 phenotypic resistance had been evident for three passages) in the different conditions and designated as follows: OS p9 (virus passaged nine times in the presence of OS alone), λ p9 (virus passaged nine times in the presence of IFN λ1 alone), OS/λ p9 (virus passaged nine times in the presence of OS plus IFN λ1), and UTR p9 (virus passaged nine times in the absence of selective drug pressure). In parallel, we also tested the A(H1N1)pdm09 Stock virus (N09 Stock) produced in MDCK cells and never passaged in Calu-3, to assess whether its sensitivity to the drugs was comparable to that of UTR p9. The dose range of OS (from 10 μM to 640 μM) was administered 1 hpi, while that of IFN λ1 (from 37 ng/mL to 1000 ng/mL) was given at 24 hbi plus at 1 hpi. After 24 h, the number of infectious viral particles released in the supernatant was determined by plaque assay in MDCK and the EC50 value was calculated in comparison to infected control wells without drug.

As expected, OS p9 was resistant to OS, with a shift in the EC50 by a factor of two orders of magnitude compared with the UTR or N09 Stock (Figure 4A). On the contrary, OS/λ p9 displayed a level of susceptibility similar to that of the UTR or N09 stock, further confirming that the resistance against OS did not emerge when the drug was co-administered with IFN λ1 (Figure 4A). All variants were sensitive to IFN λ1 in a dose-dependent manner (Figure 4B). The EC50 against UTR p9 was increased by 1.8 fold compared with the N09 stock. However, this difference was not significant and was probably determined by the viral adaptation to Calu-3 cells (see below). The EC50 λ p9 was increased by 2.5 fold compared with the N09 Stock, but also this change was not significant (Figure 4B). Lastly, there was no significant difference between λ p9 and UTR p9, at any of the tested doses of IFN λ1, suggesting that no resistance vs. IFN λ1 emerged. This finding is in line with previously published data showing that IAVs can develop resistance to type III interferons in vitro after >20 passages in Calu-3 cells [48].

### 3.4. Viral Fitness Assessment of the A(H1N1)pdm09 Variants In Vitro and Ex Vivo

The titers of the OS variant after p8 were significantly higher compared with the UTR (Figure 3A); however, gain-of-resistance is often associated with loss of viral fitness [51]. We thus determined the growth kinetics of N09 Stock, OS p9, λ p9, OS/λ p9, and UTR p9 strains at 24, 48, and 72 hpi in Calu-3 cells. We did not observe any significant difference across the variants, confirming that the acquisition of OS resistance does not result in a loss of fitness (Figure 5A).

To validate these results in a more relevant culture model, we performed another fitness assay in Mucilair. Mucilair tissues are human airway epithelia reconstituted in vitro from cells isolated from the respiratory tract of healthy donors. They consist of ciliated, goblet, and basal cells cultured at the air–liquid interface in a 3D structure. Mucilair perfectly recapitulates the barrier functions and mucociliary responses of the respiratory mucosa and it represents an ideal model to study respiratory viruses [51,52,53]. We thus infected Mucilair tissues with the same infectious doses of each variant on their apical side at 33 °C to simulate an upper respiratory infection. Then, we monitored the viral growth over time, collecting the virus released apically every 24 h. Overall, the N09 stock grew similarly to OS p9 and UTR p9 (Figure 5B), indicating that neither viral adaptation to Calu-3 cells, nor the OS resistance, affected viral fitness in Mucilair. OS/λ p9 and λ p9 displayed a similar kinetic and grew slower than the N09 Stock (*p* < 0.05 at 48, 72, and 96 hpi for both λ p9 and OS/λ p9), suggesting that IFN λ1 and the combined treatment selective pressure in Calu-3 cells impact the fitness of A(H1N1)pdm09 when grown in reconstituted human airway epithelia.

### 3.5. Sequence Analysis of Selected A(H1N1)pdm09 Variants

We next sequenced the entire genome of all the variants generated at p9, together with the N09 Stock (Table 2). The H275Y mutation, known to confer resistance to OS [54], was found in the NA segment of the variant passaged nine times in presence of OS, but not in λ p9 or OS/λ p9 (Table 2). In addition, the NA of all analyzed variants and of the original viral stock contained D344N and D354G, both permissive mutations necessary for maintaining full NA function in the presence of H275Y [55] (Table 2). H275Y decreases NA surface expression while D344N and D354G allow the viruses to tolerate this defect by altering, respectively, enzyme affinity and activity, thus restoring viral fitness [12,56]. We also sequenced earlier passages of the OS variant to document the time of occurrence of H275Y. We started to detect the mutation in the NA segment of OS p7 (but not in OS p6) as a minor quasi-species, which slightly increased at p8 to become the majority population at P9. Considering that the sensitivity of detection for Sanger sequencing is estimated to be approximately 15% to 20% of the mutant allele frequency, we estimated that OS administered as a single treatment induced the emergence of H275Y between p6 and p7. To define to which extent IFN λ1 co-administration delayed the resistance emergence, we also sequenced the NA gene of OS/λ p10, 11, and 12 (Table 2). H275Y appeared as a minor population in OS/λ p10, became more represented at p11, and then became dominant at p12, which correlates well with the viral yield of OS/λ p12 (Figure 3C). These results demonstrate that IFN λ co-administration delayed the emergence of the dominant H725Y by three passages, from p9 (in the OS single treatment condition) to p12 (in the OS/λ combined treatment condition), even in the presence of permissive mutations that facilitate its occurrence. Interestingly, we highlighted three amino acid substitutions, K62R, G239D, and Q240R, in the HA segment of UTR p9 and λ p9 variants (Table 2). All these mutations occurred in the HA globular head [57]. More specifically, Q240R lies in the HA receptor binding site and seems to be associated with a preference for alpha 2,3 rather than alpha 2,6 sialic acid receptors [58]. As they were only found in the UTR p9 and in the λp9 variants, we surmised that these mutations resulted from the adaptation of the virus to Calu-3 cells, whose surface sialic acid differs from that of the MDCK cells used to produce the stock virus [59]. We therefore investigated the speed of emergence of these HA mutations in UTR p9 and we found that they occurred upon only four passages in Calu-3 cells (Table 2). Lastly, we identified other substitutions, all present as a mixed population, in the PA, PB1, PB2, NP, and M segments of the UTR p9, λ p9, OS p9, and OS/λ p9 variants (Table 2). These mutations were not all represented in each variant, except for K57R (M segment), which probably resulted from viral adaptation to Calu-3 cells (Table 2). Moreover, to the best of our knowledge, they are either not reported in the literature, or not associated with a specific phenotype.

### 3.6. Effect of HA Changes on Receptor Specificity

To evaluate the effect of the K62R, G239D, and Q240R HA1 mutations on HA binding properties, we examined the affinity of all our A(H1N1)pdm09 variants to C11-6′, a virucidal compound exposing a residue of 6′SLN (6′sialyl-N-acetyllactosamine), which specifically binds H1N1 HA, thus inducing the inactivation of extracellular viral particles [44]. To assess the receptor specificity of HA independently of any bias introduced by the host cell, we first pre-incubated a dose range of C11-6′ with a fixed amount of virus (corresponding to a MOI of 0.1 PFU/cell). Then, we inoculated the mixed virus plus compound on MDCK cells and determined the number of infected cells by immunohistochemistry upon one cycle of viral replication. The N09 Stock, OS p9, and OSλp9, which did not bear K62R, G239D, or Q240R, were sensitive to C11-6′ to the same extent, with similar EC50 values (Figure 6). On the other hand, C11-6′ did not prevent the infection of UTR p9 and λ p9 variants, and the EC50 value against these variants was two orders of magnitude larger than the one measured against the N09 Stock (Figure 6). These data suggest that K62R, G239D, and Q240R, which emerged during repeated passages in Calu3 cells, shifted the sialic acid preference of the viruses. Of note, these mutations did not appear under the selective pressure of OS, but they occurred despite the presence of IFN λ1.

## 4. Discussion

Rapidly acquired genetic variability has made rational drug design against IVs extremely hard. The problem underlying this challenge is the inevitable development of drug resistance, where changes in a very small number of amino acid residues in the targeted viral protein are sufficient to reduce or abolish the efficacy of the drug. NAIs provide the front line of defense against IV infection and the response to the next influenza pandemic will probably rely on the extensive use of this class of antivirals, combined with other transmission control measures [62]. However, since resistant variants can arise either naturally or as a result of drug administration, it is very likely that the use of NAIs on the scale anticipated for the control of pandemic influenza will create a unique selective pressure for the emergence and spread of resistant strains [63]. There is an unmet need for new treatment regimens that can reduce the risk of resistance appearance.

Combination therapy with compounds having different mechanisms of action is a possible option for influenza treatment. In this work, we assessed the effectiveness of OS, combined with IFN λ1, against the emergence of A(H1N1)pdm09 drug-resistant variants in Calu-3 cells. We found that, in line with the literature, OS administered as a single drug effectively reduced viral replication until passage 4, but rapidly induced the emergence of H275Y in the NA gene, which started between passages 6 and 7. This mutation became dominant at passage 9 and was associated with a significantly reduced susceptibility to OS, with an increase in the EC50 by two orders of magnitude. The antiviral effect of IFN λ1 was weak, as the titers generated at each passage were substantially similar to those produced in the UTR condition and, accordingly, the drug alone did not induce the emergence of resistance after twelve passages. Nonetheless, when OS was co-administered with IFN λ1, H275Y did not appear at p7 but emerged only at p10. The OS/λ p9 was susceptible to OS to the same extent as the N09 Stock, UTR p9, and λ p9 variants. H275Y hinders viral replication and its spread is made possible by several compensatory mutations that restore viral fitness [55]. Two of these amino acid substitutions, D344N and D354G, were present in all our variants, including the stock virus. D344N was identified as a major determinant of increased NA affinity for sialic acids, while D354G increases the activity of the enzyme and can compensate alone for the remaining H275Y-generated functional defects [55,64]. Both mutations are present in currently circulating H1N1 strains. This result emphasizes the delayed emergence of OS resistance in the presence of IFN λ1 co-administration, even in strains that are tolerant to H275Y. Importantly, H275Y is the major but not the only mutation known to confer resistance to OS. I223V and S247N have also been observed alone or in combination with H275Y [65] in the NA segment of resistant 2009 pandemic H1N1 viruses, but none of these additional substitutions appeared in the OS p9 variant.

OS resistance in the H1N1pdm09 results in an increased viral infectiousness [66]. Of note, in Calu-3 cells, we did not observe any difference in the kinetic of viral replication between the variants generated in the same cell line and the N09 Stock produced in MDCK cells and OS resistance did not impact viral fitness. Similarly, in ex vivo reconstituted respiratory tissues, OS p9, UTR p9, and the N09 Stock replicated with a similar kinetic, but significantly faster compared with λ p9 and OS/λ p9. This suggests that OS resistance does not impact viral fitness in respiratory tissues.

To monitor the emergence of amino acid substitutions induced by the adaptation of the WT virus to growth in Calu-3 cells, we passaged the parental virus in parallel in the absence of any antiviral selective pressure. Three HA mutations, K62R, G239D, and Q240R, were found in the HA globular head [57] of UTR p9 and λ p9 variants and resulted in a reduced binding affinity for the 6′SLN-thrisaccharide, as shown by the lack of susceptibility towards C11-6′ [29,44]. These amino acid substitutions are a consequence of the viral adaptation to Calu-3 cells and occur upon four passages in this cell line. It is not clear why these mutations did not confer a proliferative advantage in Calu-3 cells over the N09 Stock and why they did not affect viral fitness in Mucilair. Additionally, none of them appeared in the OS or the OS/λ p9 variant, probably due to the OS selective pressure limiting viral replication and emergence of resistance. This finding poses the issue of the choice of the in vitro model when evaluating the effectiveness of an antiviral. The occurrence of K62R, G239D, and Q240R in Calu-3 does not bias the assessment of IV susceptibility to NAI, but it might jeopardize the study of other classes of antivirals, namely those targeting the HA receptor binding domain. Sequencing analysis detected additional non-conservative and previously undescribed mutations in PA (V100L and N65K), PB2 (T303S and I57M), NP (P318Q), and M (K57R) segments of the variants generated at p9. Besides K57R, these mutations were not equally distributed across the viral strains. Nonetheless, their presence and distribution could not explain the differences in viral fitness in Mucilair between the N09 Stock and the two variants OS/λ p9 and λ p9. Sequencing tools more accurate than the Sanger method, such as NGS technologies, would be needed to better profile the genetic features of these viral strains and the association with different viral fitness ex vivo.

The findings presented in this study support the prospective therapeutic application of a combined OS/IFN λ1 treatment against influenza infection. IFN λ1 impaired viral replication at multiple levels, thus preventing the occurrence of H275Y. In our settings, the combinatorial effect of the two compounds was investigated upon twelve passages in the presence of drugs. Additional studies will be needed to precisely estimate to which extent IFN λ1 can delay the emergence of OS resistance in vivo. Calu-3 cells respond to IV infection by inducing an IFN response [46], which is however not sufficient to prevent the emergence of OS-resistant strains. Studies in B6.A2G-Mx1 mice demonstrated that IFN λ, administered intranasally and upon infection, is a potent anti-influenza therapeutic without the inflammatory side effects of IFN α treatment, suggesting IFN λ as a potential treatment of choice against IV [32,38,67]. Further in vivo studies will be necessary to investigate whether IFN λ will be synergistic with OS and delay the emergence of resistance, thus extending the OS treatment window. In addition, similar studies combining IFN λ with other anti-IV antivirals known to induce viral resistance, such as baloxavir [68], should be performed to find out whether a similar delay in resistance emergence is observed. Another lesson that can be drawn from our work is that caution should be exercised when selecting in vitro models to test antivirals against IV.

## Figures and Tables

**Figure 1 microorganisms-09-01196-f001:**
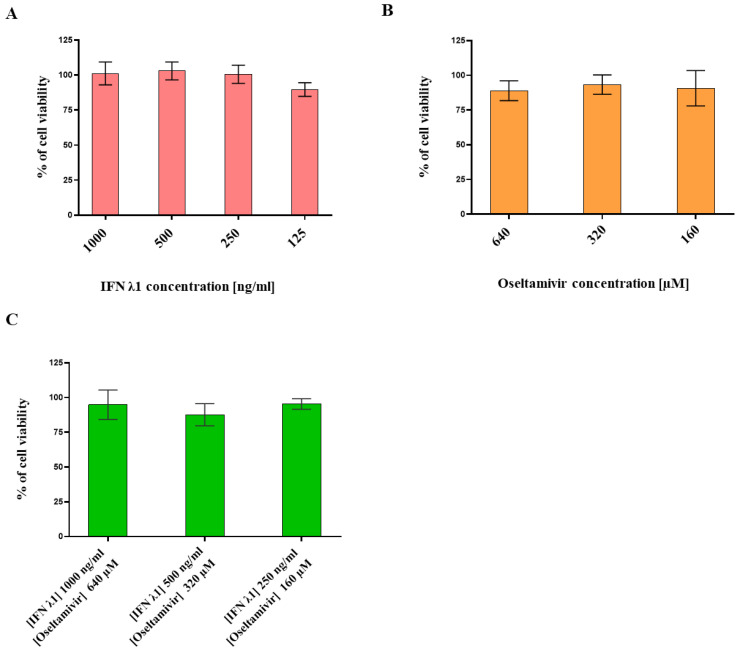
Viability assessment of Calu-3 cells grown in the presence of a dose range of IFN λ1 and OS administered alone or in combination. IFN λ1 treatment (**A**), OS treatment (**B**), IFN λ1 plus OS co-treatment (**C**). Cell viability was measured by MTT assay. The % of cell viability was calculated based on an untreated control at 48 h post treatment. The results represent the mean and standard deviation from two independent experiments performed in duplicate. IFN λ1 = human IFN λ1; OS = Oseltamivir.

**Figure 2 microorganisms-09-01196-f002:**
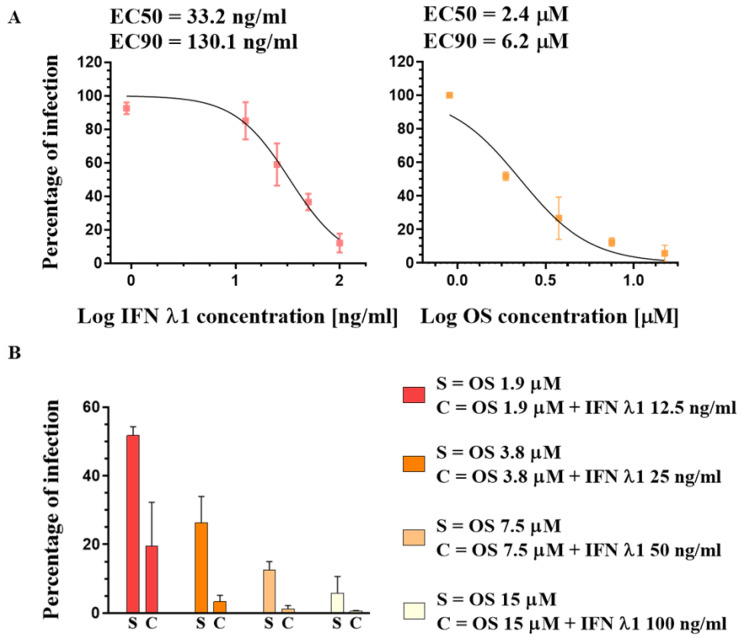
Comparison of IFN λ1 and OS antiviral activity against A(H1N1)pdm09 in Calu-3 cells. (**A**) Dose–response curve of the single treatments. The percentages of infection represent the number of plaques induced by infectious supernatant collected at 24 hpi, in treated versus untreated controls. The infection was performed with a MOI of 0.1 PFU/cell. EC50 = Half maximal effective concentration; EC90 = drug concentration at which 90% of the maximal effect is observed. (**B**) Bar plot showing the percentages of infection (determined as in A) of a dose range of OS administered individually (S) or of a dose range of OS administered together with a dose range of IFN λ1 (C). The results represent the mean and standard deviation from two independent experiments performed in duplicate.

**Figure 3 microorganisms-09-01196-f003:**
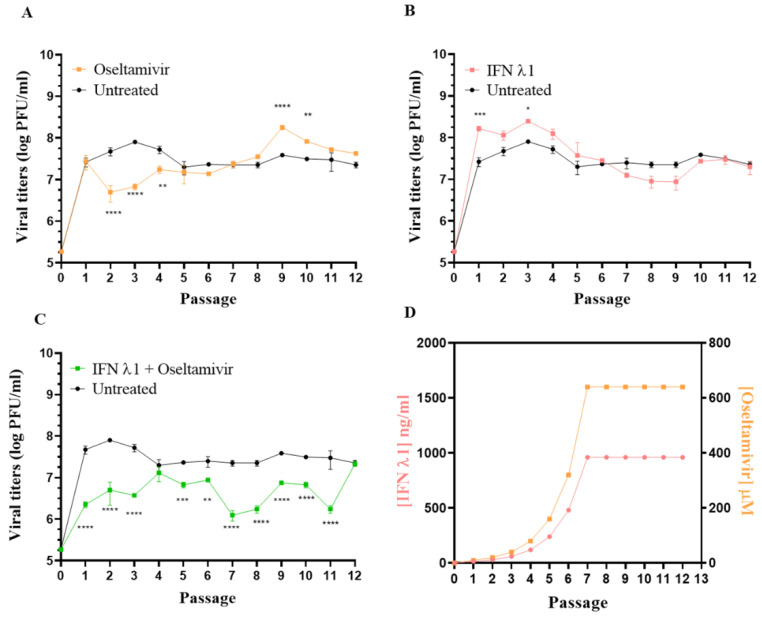
Comparison of viral titers from viral supernatants collected from Calu-3 cells infected with A(H1N1)pdm09 for 48 h in the presence of increasing concentrations of OS (**A**), IFN λ1 (**B**), and OS plus IFN λ1 (**C**), this over 12 consecutive passages in Calu-3 cells. Each population is shown together with the untreated control, serially passaged in the absence of selective drug pressure. The titer (performed in MDCK cells) is expressed in plaque-forming units per milliliter (log PFU/mL). The drug concentration at each passage is also shown (**D**). The results represent the mean and standard deviation from two independent titrations. *, *p* ≤ 0.05; **, *p* ≤ 0.01; ***, *p* ≤ 0.001; ****, *p* ≤ 0.0001.

**Figure 4 microorganisms-09-01196-f004:**
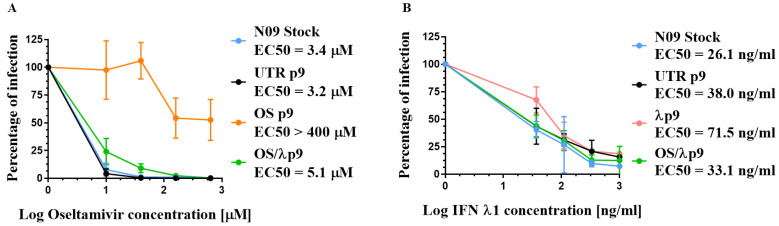
Antiviral effect of OS (**A**) and IFN λ1 (**B**) against the A(H1N1)pdm09 variants grown for nine passages in Calu-3 cells and measured by a virus yield reduction assay in MDCK cells. Mean EC50 values and standard deviations were calculated from two independent experiments performed in duplicate. OS p9 = virus passaged nine times in the presence of OS alone; λ p9 = virus passaged nine times in the presence of IFN λ1 alone; OS/λ p9 = virus passaged nine times in the presence of OS plus IFN λ1; UTR p9 = virus passaged nine times in the absence of selective drug pressure; N09 Stock = virus produced in MDCK cells and never passaged in Calu-3.

**Figure 5 microorganisms-09-01196-f005:**
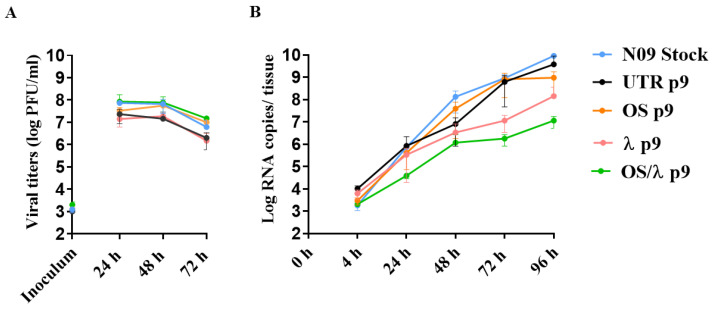
Replication of the A(H1N1)pdm09 variants in Calu-3 cells (**A**) and in ex vivo human airway epithelia reconstituted at the air–liquid interface (**B**). In (**A**) Calu-3 cells were infected with a MOI of 0.01 PFU/cell and viral titers were measured by plaque assay from daily collected cell supernatants. At each time point, the entire infectious supernatant was collected and new medium was added, thus allowing for the measurement of daily viral production. In (**B**), Mucilair respiratory tissues were infected apically with 5 × 10^4^ PFU, determined by plaque assay in MDCK cells, per tissue. Viral loads, expressed in RNA copies/tissue, were quantified by RT-qPCR from daily collected apical. The residual virus present on the apical side of the tissues upon the inoculum removal, at 4 hpi, was also quantified by RT-qPCR. Statistical significance described in the text is not shown in the figure for clarity. Abbreviations as in Figure 4. The results represent the mean and standard deviation from two independent experiments performed in duplicate.

**Figure 6 microorganisms-09-01196-f006:**
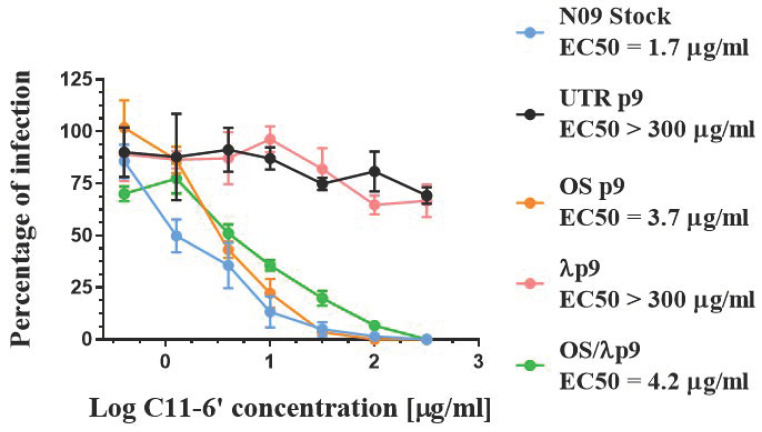
Dose–response curves demonstrating the antiviral activity of C11-6′ when pre-incubated with A(H1N1)pdm09 variants before infection in MDCK cells. OS p9 = virus passaged nine times in the presence of OS alone; λ p9 = virus passaged nine times in the presence of IFN λ1 alone; OS/λ p9 = virus passaged nine times in the presence of OS plus IFN λ1; UTR p9 = virus passaged nine times in the absence of selective drug pressure; N09 Stock = virus produced in MDCK cells and never passaged in Calu-3. Mean EC50 values and standard deviations were calculated from two independent experiments performed in duplicate.

**Table 1 microorganisms-09-01196-t001:** Primers used to amplify gene segments of A(H1N1)pdm09 viruses.

Gene	Primer	Binding Position	Sequence *
HA	Forward 1	1–22	TGT AAA ACG ACG GCC AGT ATGAAGGCAATACTAGTAGTTCTG
HA	Reverse 1	901–921	CAG GAA ACA GCT ATG ACC GAGGCTGGTGTTTATAGCACC
HA	Forward 2	802–822	TGT AAA ACG ACG GCC AGT CCGAGATATGCATTCGCAATG
HA	Reverse 2	1676–end	CAG GAA ACA GCT ATG ACC TTAAATACATATTCTACACTGTAGAG
NA	Forward 1	1–26	TGT AAA ACG ACG GCC AGT ATGAATCCAAACCAAAAGATAATAAC
NA	Reverse 1	823–842	CAG GAA ACA GCT ATG ACC CAGGAGCATTCCTCATAGTG
NA	Forward 2	706–729	TGT AAA ACG ACG GCC AGT GGTTCTTGCTTTACTGTAATGACC
NA	Reverse 2	1386–1410	CAG GAA ACA GCT ATG ACC TTACTTGTCAATGGTAAATGGCAAC
NA ^●^	Forward	727–748	CTGTAATGACCGATGGACCAAG
NA ^●^	Reverse	913–935	CAGATTCTGGTTGAAAGACACCC
PB1	Forward 1	1–22	TGT AAA ACG ACG GCC AGT ATGGATGTCAATCCGACTCTAC
PB1	Reverse 1	996–1017	CAG GAA ACA GCT ATG ACC CATGCTCAGGATGTTTCTGAAC
PB1	Forward 2	597–618	TGT AAA ACG ACG GCC AGT GGTCACGCAAAGAACAATAGG
PB1	Reverse 2	1521–1542	CAG GAA ACA GCT ATG ACC CACTCCAAAGCTGGGTAGCT
PB1	Forward 3	1301–1321	TGT AAA ACG ACG GCC AGT CAATATACTGGTGGGATGGGC
PB1	Reverse 3	2251–end	CAG GAA ACA GCT ATG ACC TTATTTTTGCCGTCTGAGTTCTTC
PB2	Forward 1	1–23	TGT AAA ACG ACG GCC AGT ATGGAGAGAATAAAAGAACTGAGAG
PB2	Reverse 1	998–021	CAG GAA ACA GCT ATG ACC CTTTCTTGACTGATGATCCGC
PB2	Forward 2	603–626	TGT AAA ACG ACG GCC AGT GGTGGCGTACATGCTAGAAAG
PB2	Reverse 2	1552–1573	CAG GAA ACA GCT ATG ACC CAGTTCCTTGCGTTTCACTGAC
PB2	Forward 3	1230–1248	TGT AAA ACG ACG GCC AGT GATCAAGGCAGTTAGGGGC
PB2	Reverse 3	2258–end	CAG GAA ACA GCT ATG ACC CTAATTGATGGCCATCCGAATTC
PA	Forward 1	1–23	TGT AAA ACG ACG GCC AGT ATGGAAGACTTTGTGCGACAATG
PA	Reverse 1	996–1017	CAG GAA ACA GCT ATG ACC CTTCCAAGCCATGAGGTAATTG
PA	Forward 2	616–36	TGT AAA ACG ACG GCC AGT GAGATTACAGGAACTATGCGC
PA	Reverse 2	1519–1540	CAG GAA ACA GCT ATG ACC CATTTCTCAAATGAGACCTTCC
PA	Forward 3	1162–1184	TGT AAA ACG ACG GCC AGT GGAGACCTTAAACAGTATGACAG
PA	Reverse 3	2130–end	CAG GAA ACA GCT ATG ACC CTACTTCAGTGCATGTGTGAGG
M	Forward	1–21	TGT AAA ACG ACG GCC AGT ATGAGTCTTCTAACCGAGGTC
M	Reverse	959–982	CAG GAA ACA GCT ATG ACC TTACTCTAGCTCTATGTTGACAAA
NP	Forward 1	1–20	TGT AAA ACG ACG GCC AGT ATGGCGTCTCAAGGCACC
NP	Reverse 1	801–821	CAG GAA ACA GCT ATG ACC GATTTATGTGCAACTGATCCC
NP	Forward 2	702–721	TGT AAA ACG ACG GCC AGT CCAGAGGGCAATGATGGATC
NP	Reverse 2	1477–end	CAG GAA ACA GCT ATG ACC TCAACTGTCATACTCCTCTGC
NS	Forward	1–20	TGT AAA ACG ACG GCC AGT ATGGACTCCAACACCATGTC
NS	Reverse	840–863	CAG GAA ACA GCT ATG ACC GTAGAAACAAGGGTGTTTTTTATC

* The sequence of the M13 tail is indicated in blue. ^●^ Primers used for NA sequencing.

**Table 2 microorganisms-09-01196-t002:** Amino acid changes identified by sanger sequencing in selected A(H1N1)pdm09 variants.

Virus	Passage No.	Oseltamivir (μM)	IFN λ1 (ng/mL)	HA *	NA **	PA	PB1	PB2	NP	M
N09 Stock	-	-	-	-	D344N, D354G	-	-	-	-	-
UTR p3	3	-	-	-	D344N, D354G	ns	ns	ns	ns	ns
UTR p4	4	-	-	K62R, G239D, Q240R	D344N, D354G	ns	ns	ns	ns	ns
UTR p9	9	-	-	K62R, G239D, Q240R	D344N, D354G	V100L ^■^	A1920T °	T303S ^■^	-	K57R ^■^
OS p7	7	640	-	ns	H275Y ^■^, D344N, D354G	ns	ns	ns	ns	ns
OS p8	8	640	-	ns	H275Y ^■^, D344N, D354G	ns	ns	ns	ns	ns
OS p9	9	640	-	-	H275Y, D344N, D354G	N65K ^■^	-	-	-	K57R ^■^
λ p9	9	-	960	K62R, G239D, Q240R	D344N, D354G	-	-	I57M ^■^	P318Q ^■^	K57R ^■^
OS/λ p9	9	640	960	-	D344N, D354G	-	-	-	-	K57R ^■^
OS/λ p10	10	640	960	ns	H275Y ^■^, D344N, D354G	ns	ns	ns	ns	ns
OS/λ p11	11	640	960	ns	H275Y ^■^, D344N, D354G	ns	ns	ns	ns	ns
OS/λ p12	12	640	960	ns	H275Y ^■^, D344N, D354G	ns	ns	ns	ns	ns

* Based on H1 numbering as in [60]. ** Based on N1 numbering as in [61]. ^■^ Present as a mixed population. ° Silent mutation, for which the nucleotide substitution is shown. ns = not sequenced.

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
