# Peer review of "Interferon Lambda Delays the Emergence of Influenza Virus Resistance to Oseltamivir"

_microorganisms, 2021, doi:10.3390/microorganisms9061196_

Round 1
Reviewer 1 Report
Please see below for specific comments:
Figure1: What is the data that led the authors to their conclusion that "no effect on the metabolic activity of Calu-3 cells was detected".
Rest of the manuscript is well presented and written within the scope of the topic and chosen journal.
Author Response
Response to Reviewer 1 Comments
Figure1: What is the data that led the authors to their conclusion that "no effect on the metabolic activity of Calu-3 cells was detected”. Rest of the manuscript is well presented and written within the scope of the topic and chosen journal.
We thank the reviewer for his/her comments and appreciation of our work. We concluded that "no effect on the metabolic activity of Calu-3 cells was detected" because we performed a viability assay that uses the reduction of a yellow tetrazolium salt (3-(4,5-dimethylthiazol-2-yl)-2,5- diphenyltetrazolium bromide, or MTT) to measure cellular metabolic activity as a proxy for cell viability.
We agree with the reviewer that this statement, in its original declination, is not entirely accurate nor particularly helpful. In the revised manuscript we replaced the statement with the following "no effect on the viability of Calu-3 cells was detected" (line 252).
Reviewer 2 Report
The authors reported the effect of combinational treatment with oseltamivir (OS) and interferon lambda to delay the emergence of influenza virus resistant to oseltamivir. In this study, the authors utilized in vitro infection model using Calu-3, a submucosal gland cell line, which was generated from a bronchial adenocarcinoma, and monitored the emergence of drug-resistant mutant viruses over 11 passages. H275Y, a mutant of neuraminidase, which only appeared in OS treated p9 culture, was not detected till p11, when treated with interferon lambda in Calu-3 cells. Since It is well established that combinational therapy with different mechanisms of actions reduces the emergence of drug resistant mutants, the results in this study is not entirely novel, rather expected. There are some points that need to be addressed to improve the quality of the manuscript.
Major points
- As shown in the case of HA mutants following no treatment or interferon lambda treatment, the variants in different cell lines may be different. It is necessary to see whether the authors can reproduce the similar results in MDCK cells, compared to those in Calu-3 cells.
- The authors said that IFN-lambda “delays” the emergence of influenza virus resistance to oseltamivir, but the delayed emergence of resistant virus is not shown. How many more passages would the authors repeat before the H275Y mutant appears in the presence of OS and IFN-lambda?
Reviewer 3 Report
In the present manuscript, Medaglia, Zwygart and coll. evaluate in an in vitro and ex vivo model of influenza H1N1 infection the potential effect of a combined treatment with recombinant IFNL1 and NAI inhibitor oseltamivir on the emergence of antiviral resistance to the latter. The manuscript is well written and intends to answer a pertinent question in terms of the management of influenza antiviral resistance. The authors conclude that “… the co-administration of interferon lambda drastically delayed the emergence of drug-resistant influenza virus variants”, however, in my opinion, the study has major methodological issues that might bias data interpretation. My recommendation is that the points detailed below must be addressed before considering the study for potential publication.
Major concerns:
- Section 3.2:
- Is there a valid reason for determining EC50 values at 24hpi while the readout of all other experiments presented in the manuscript is 48hpi? Please determine EC50 at 48hpi. Moreover, according to Table 2, calculated EC50 for both IFNL1 and OS do not fall within the dose-range tested. Please extend the dose-range to lower values and provide dose-response curves to, among other things, estimate the difference between EC50 and EC90 values. Finally, the statement “IFN λ1 exerts an antiviral effect only when administered in pre- and post-treatment in cell lines” is not accurate. In fact, reference 48 uses pre-treatment only and, to my knowledge, no evidence is provided to affirm that post infection treatment with IFNL1 is not effective.
- Section 3.3:
- The authors state that serial passages start at the EC50 concentration but it seems that the starting doses (10 uM OS and 37 ng/ml IFNL1) do not match this affirmation. Please explain and correct/repeat as necessary. Minor point: EC50 values were determined in Calu-3 instead of MDCK; please correct the text.
- I consider the affirmation that the difference in viral titers at p4 in Fig 2A indicates “a reduced susceptibility of the viruses to the drug” is an overstatement. Differences observed in previous passages are <1log, which considering the inherent variability of the PFU technique as well as the reduced number of biologic replicates seems like very weak “evidence” of viral reduction, despite the results of the unpaired t test. Moreover, the “suggested emergence of an OS resistant variant” at p4-p5 is not proved in the study. The same applies for the statement concerning the increased viral fitness of the resistant variant at p9-p10, which is moreover not supported by observable differences in plaque shape or size.
- Fig 2B: the doses of IFNL1 do not seem to induce an antiviral effect, which challenges the validity of the experiment as well as the affirmation that “no resistance to vs IFNL1 emerged” and further highlights the need of showing the dose-response curves for the single drugs and combination used in this study.
- Fig 2C: as mentioned above, I do not agree it would be valid to affirm that antiviral resistance emerged at p5 in the OS group and use that as a “strong suggestion” of the effect of combination on the delay of resistance emergence, notably in the face of sequencing results (Table 3) that do not support the p5 hypothesis. Minor: please revise stats, notably P values from p3 compared to p5.
- Section 3.4:
- While I appreciate the use of the Mucilair model, it is really unfortunate that fitness experiments were performed using viral inoculum titrated in RT-qPCR and not in PFU like the rest of the manuscript. This completely abrogates a meaningful comparison of viral fitness results between the two models and even within the ex vivo experiment, since differences in genome copies do not necessarily imply the same differences in infectious particles.
- Section 3.5:
- The absence of OS resistance mutations in the OS-treated arm between p8 and p9 definitely contradicts the previous statement of resistance emergence at p4-p5. Moreover, since the 275Y mutation emerges at p9 and the experiment stops at p10, I do not believe it is possible to reject the null hypothesis and affirm that the absence of the 275Y mutation at p10 in the combination group (only 1 passage later than OS) is solid proof of the effect of the combination on delaying antiviral resistance.
- Please provide complete sequence data from at least NA and HA (but preferably from all segments) of all sequenced passages to assess the potential presence of other permissive, cell adaptation or unexpected mutations that might bias the interpretation of results. For example, the described HA mutations present at different passages in some but not all viruses challenge previous affirmations on the effect of treatment on viral fitness.
Minor concerns:
- Methods: please indicate how virus samples (supernatants) were managed for the serial passages experiment. Were they freezed-thawed before infection of the next passage? If yes, was the effect of this freezing-thawing on viral titer assessed?
- Discussion: please move the phrase “Studies in B6.A2G-Mx1 mice……. of choice against IV.” to the introduction
